# A New Disease Caused by an Unidentified Etiological Agent Affects European Salamanders

**DOI:** 10.3390/ani12060696

**Published:** 2022-03-10

**Authors:** Raoul Manenti, Silvia Mercurio, Andrea Melotto, Benedetta Barzaghi, Sara Epis, Marco Tecilla, Roberta Pennati, Giorgio Ulisse Scarì, Gentile Francesco Ficetola

**Affiliations:** 1Department of Environmental Science and Policy, University of Milano, 20133 Milano, Italy; benedetta.barzaghi@unimi.it (B.B.); roberta.pennati@unimi.it (R.P.); francesco.ficetola@gmail.com (G.F.F.); 2Centre of Excellence for Invasion Biology, Department of Botany and Zoology, Stellenbosch University, Stellenbosch 7602, South Africa; mel8@hotmail.it; 3Department of Biosciences, University of Milano, 20133 Milano, Italy; sara.epis@unimi.it (S.E.); marco.tecilla@gmail.com (M.T.); giorgio.scari@unimi.it (G.U.S.); 4Laboratoire d’Ecologie Alpine (LECA), University Grenoble Alpes, CNRS, 38400 Grenoble, France; 5Laboratoire d’Ecologie Alpine (LECA), University Savoie Mont Blanc, CNRS, 38400 Grenoble, France

**Keywords:** protist, Mesomycetozoea, pathogenic, amphibians, fire salamander, *Salamandra salamandra*

## Abstract

**Simple Summary:**

In the last few years, multiple new infectious diseases have affected amphibians, causing unprecedented declines and extinctions at the global scale. Many of these diseases are caused by pathogens that have been described during recent decades. In this study we report a novel disease, affecting European amphibians. In a protected area of Northern Italy, since the autumn of 2013, we started to find some adult salamanders with cysts at the throat level (subgular region). Following this observation, we ran a regular monitoring of the salamander population and performed multiple morphological and molecular investigations to identify the cause of this undescribed disease. The cysts surround peculiar cells, probably protists, of about 30 µm covered by numerous motile cilia/undulipodia. Despite multiple attempts with a broad spectrum of techniques, the detailed identification remains challenging as we have been unable to match its features with previously described organisms. We provide the results achieved till now to promote a rapid dissemination on this new enigmatic wildlife pathogen and to create a basis for further and deeper studies.

**Abstract:**

New pathologies are causing dramatic declines and extinctions of multiple amphibian species. In 2013, in one fire salamander population of Northern Italy, we found individuals with undescribed cysts at the throat level, a malady whose existence has not previously been reported in amphibians. With the aim of describing this novel disease, we performed repeated field surveys to assess the frequency of affected salamanders from 2014 to 2020, and integrated morphological, histological, and molecular analyses to identify the pathogen. The novel disease affected up to 22% of salamanders of the study population and started spreading to nearby populations. Cysts are formed by mucus surrounding protist-like cells about 30 µm long, characterized by numerous cilia/undulipodia. Morphological and genetic analyses did not yield a clear match with described organisms. The existence of this pathogen calls for the implementation of biosecurity protocols and more studies on the dynamics of transmission and the impact on wild populations.

## 1. Introduction


**Epigraph**


“Нo как же этo так? Ведь этo же чудoвищнo! Этo чудoвищнo, гoспoда”, -пoвтoрил oн, oбращаясь к жабам в террарии, нo жабы спали и ничегo ему не oтветили.” (“But how can it be? It’s monstrous! Quite monstrous, gentlemen,” he repeated, addressing the toads in the terrarium, who were asleep and made no reply).M. Bulgakov, Рoкoвые яйца

The emergence and spreading of novel diseases in wildlife is one of the most challenging threats to both biodiversity and human well-being [1,2,3,4]. Emerging wildlife diseases can quickly have profound broad-scale impacts, including ecological disturbance, economic and agricultural impacts, and even human losses [4,5]. The rapid detection of novel wildlife diseases is thus crucial, and often requires an interdisciplinary approach, including for instance morphological and genetic investigations, ecological assessments, and management consultations [5,6,7]. However, when diseases emerge in wildlife organisms, their detection is not easy and some pathogens may remain unnoticed till their spreading becomes difficult to control [8,9]. This could be particularly true for elusive or poorly studied animals, as ‘non-mammals’ vertebrates, and when pathologies are strongly different from the already known diseases.

Amphibians provide a clear example of the damage that the increasing spread of novel diseases may play over short periods to biodiversity at a global scale. In the last decades, recently discovered pathogens have caused dramatic declines and extinctions of populations and even species of amphibians [10,11]. The most studied pathologies threatening amphibians are chytridiomycoses caused by the fungi *Batrachochytrium salamandrivorans* and *Batrachochytrium*
*dendrobatidis* [12,13,14], which are involved in the decline of >500 amphibian species and determine the greatest loss of wildlife biodiversity linked to a pathogen [13]. Amphibians are also susceptible to viruses such as *Ranavirus* sp. [15], which is the main infectious pathogen of multiple aquatic ectothermic vertebrates and is often implicated in mass die-offs of amphibians [16,17], even causing 100% mortality in tadpoles of some anuran species [18,19]. Moreover, amphibians also host understudied pathogens such as the fungal-like protists Mesomycetozoea [20,21,22,23]. Their biological features are typical of pathogens able of determining emerging infectious diseases in different amphibians as in the case of the genus *Amphibiocistydium* [24,25], but information on these organisms and their impacts remains scanty.

With this paper, we want to bring to the attention of the world’s scientific community a novel disease affecting a widespread European amphibian, the fire salamander (*Salamandra salamandra* Linnaeus, 1758). In 2013, two individuals with an odd cyst at the level of the throat (Figure 1) were discovered in a small, protected area of Northern Italy. During following surveys, we observed similar cysts in multiple individuals. Therefore, we directed great effort to identify the cause of the cysts and to understand the extent of their spreading. In particular, we monitored the occurrence of the cysts during the following years and performed detailed morphological, ultrastructural and molecular analyses in order to characterize their content. We believe this information must be disseminated across scientists and managers before this phenomenon strikes other amphibian populations. 

## 2. Materials and Methods

### 2.1. The Fire Salamander: Study Area and Monitoring

The fire salamander is ovoviviparous and widespread in Europe; this species mainly inhabits hilly landscapes covered by broadleaf forests and breeds in streams and other freshwater sites [26]. When adult, fire salamanders show strong terrestrial habits, with dispersal abilities range from 200 to 1300 m (but generally up to 500 m), and nocturnal behaviour [27]. Although widespread, the species has recently suffered catastrophic declines in the northern part of its range because of the chytrid fungus *Batrachochytrium salamandrivorans* [14]. Our study focuses on the regional protected reserve named “Fontana del Guercio” located around 30 km North of the city of Milan (NW Italy) and that is recorded as a European Site of Community Interest. The area is characterized by an extended broadleaf wood crossed by a slow flowing stream and numerous springs where the fire salamander breeds. This area has undergone annual monitoring since 2010. Since 2014, after the observation of the first two individuals bearing the subgular cysts, till 2020, we established yearly surveys with three transects covering the main area of the reserve (Figure 2). From 2016, to monitor the spreading of the pathogen occurrence in the surroundings, we also sampled three additional transects in a nearby (1.6 km distant) wooded area (named “Olgelasca”) occurring in a parallel valley and connected only through some wooded stretches Transects are on average ± SE 462.7 ± 95.7 m long and 18.7 ± 2.6 m wide; they follow the main paths and cover all the terrestrial habitats surrounding multiple breeding sites of the fire salamander. We performed at least three-night surveys in each autumn season (October–December) for each transect. On each survey, we collected all the salamanders we encountered; after recording the position with a GPS, we checked each individual for the occurrence of the pathogen, and then we photographed on millimetre paper and weighed it, before releasing it in the same place of collection.

The occurrence of the cyst was detected by examining the throat of the individuals. If at least one cyst was visible, we considered the salamander as affected; we considered swellings larger than 2 mm of thickness, paying attention to the fact that the throat of the fire salamanders often shows small plicas that can resemble small bumps. In cases where a second swelling occurred, we recorded it. To prevent spreading the disease during surveys we handled the salamanders with nitrile gloves that we changed for each individual; each salamander was placed for weighing and picturing on a disposable bag that was then immediately disposed of. At the end of each survey, all the material used, including shoes, was disinfected with 10% bleach.

### 2.2. Cyst Extraction

To understand the origin of the cyst under the skin, a few infected animals were collected during sampling and we extracted cyst samples in vivo from salamanders’ throat using a non-invasive surgical protocol. Prior to surgery, the animals were anesthetized by a subcutaneous injection of tricaine/carbocaine 4 mg/mL (0.32 mL/g of weight). Loss of superficial reflexes was tested prior to the incision. The samples were removed by using a 0.3 cm, sterile, disposable biopsy punch with plugger (Bioseb lab, Vitrolles, France) and immediately processed for subsequent analyses (see below). After complete remission, usually occurring within 24 h in ozonized water, the salamanders were reintroduced in their collection site. In total we extracted samples of 15 cysts from 8 individuals from 2014 to 2018.

### 2.3. Cyst Cell Isolation

Under aseptic conditions, two cysts were surgically removed and immediately processed for in vitro analysis. Samples were dissected into pieces using sterile fine-tipped tweezers and disaggregated in sterile amphibian Ringer’s solution. Then, cells were centrifuged at low speed (1200 rpm/9 g) and resuspended in Medium 199 with 10% foetal bovine serum and 50 mg/L gentamicin (Sigma, Igea Marina RN, Italy). Cells were cultured in petri dishes at 12 °C for 20 days. All cultures were observed daily using an inverted phase contrast microscope. Replacement of 50% of the culture medium was carried out every day.

### 2.4. Histology

Three cysts were processed for light microscopy analysis. Samples were immediately fixed with Bouin’s solution (picric acid, formaldehyde, acetic acid, 75:25:5) for at least 24 h and rinsed several times in tap water until all the fixative solution was completely removed. Samples were then dehydrated with an ethanol series (70%, 90%, 95% and 100%), cleared in xylene and left overnight in a solution of xylene and paraffin wax 56–58 °C (1:1). Samples were then immersed in three changes of paraffin wax and finally embedded. Sections 5–7 µm thick were cut with a standard microtome and stained with Haematoxylin and Eosin (HE).

### 2.5. Electron Microscopy

To better describe cyst structure and the cells trapped inside, electron microscopy was also performed. Samples were pre-fixed with 2% glutaraldehyde in 0.1 M cacodylate buffer for two hours and, after overnight washing in the same buffer, post-fixed with 1% solution of OsO4 in 0.1 M cacodylate buffer. After standard dehydration in ethanol series (25%, 70%, 90%, and 100%), samples were washed in propylene oxide and embedded in Epon-Araldite 812 resin (Bio Optica, Milan, Italy). Semi-thin (about 0.9 μm) and ultra-thin (70 nm) sections were cut with a Reichert–Jung ULTRACUT E using glass knives. Semi-thin sections were stained with crystal violet and basic fuchsine, mounted with Eukitt (Bio Optica, Milan, Italy) and observed under a Leica light microscope. Ultrathin sections were mounted on copper grids and stained with uranyl acetate and lead citrate for electron microscopy, then observed and photographed in a Talos L120C TEM microscope.

### 2.6. Scanning Electron Microscopy

Cyst samples were immediately fixed in 1% glutaraldehyde and 0.4% formaldehyde in 0.1 M sodium cacodylate buffer (pH 7.2, 300–400 mOsm) for 30 min at 4 °C. Then, samples were washed 3 times in 0.1 M sodium cacodylate buffer over 5 min and post fixed with 1% OsO4 in 0.1 M sodium cacodylate buffer for 30 min. Cysts were washed again 3 times, dehydrated in ethanol series (50%, 70%, 80%, 90%, 96%, and absolute alcohol), and critical point dried [CPD] (Balzers CPD 030 Critical Point Dryer; Bal-Tec AG, Balzers, Liechtenstein) in carbon dioxide. Dried samples were mounted on stabs with carbon adhesive discs, then Au sputtered using a Bal-Tec SCD 050 Sputter Coater (Bal-Tec AG, Balzers, Liechtenstein), and examined with a scanning electron microscope (LEO-1430).

### 2.7. Genetic Analyses

Standard protocol for genomic DNA extraction with proteinase K from whole cyst samples was performed with some modifications. Briefly, overnight proteinase K digestion was preceded with triple liquid nitrogen/water bath in 100 °C for 2 min to guarantee effective genomic extraction [28]. Based on literature we selected different primers for DNA amplification. A pair of universal non-metazoan primers, 18S-EUK581-F (5′-GTGCCAGCAGCCGCG-3′) and 18S-EUK1134-R (5′-TTTAARKTTCAGCGCTTGSG-3′), were chosen to amplify a 544 bp fragment of 18S rDNA from protists without getting animal DNA [29]. Two protozoa-specific forward primers (P-SSU-342f and PR900f) in combination with a protozoa-specific reverse primer (PR900r) or an eukarya-specific reverse primer were used for targeting 18S rDNA gene. Primers sequences were: PR900f (5′-TTTCGATGGTAGATTGGAC-3′), PR900r (5′-CTTGTTACGACTTCTCCTTCC-3′) amplifying a 900 bp fragment [30,31]); P-SSU-342f (5′-CTTTCGATGGTAGTGTATTGGACTAC-3′) and Medlin B (5′-TGATCCTTCTGCAGGTTCACCTAC-3′) amplifying a 1360 bp fragment [32].

A second attempt to identify the host trapped in the salamander’s cyst consisted of extracting DNA directly from the cells isolated for in vitro analysis. We selected the ciliated cells occurring in the cyst using 20 µL of TE Buffer (Tris-EDTA, 100× Solution, pH 8.0) plus 3.5 µL of liquid cystic (about 10–20 cells). We performed DNA extraction using a Proteinase K protocol. All the reagents were sterile. 2 µL of PK 20 mg/mL were added to the sample mixing the solution very well. Samples were incubated in the Thermomixer at 56 °C for two hours, then at 95 °C for 5 min and finally centrifuged at 22 °C for 10 min at 20,000× *g*. The small subunit ribosomal RNA (SSrRNA) genes were amplified with the universal eukaryotic primers forward 18S F9 [59-CTG GTT GATCCT GCC AG-39] [33] and reverse 18SR1513 Hypo [59-TGA TCC TTC (CT)GC AGG TTC-39] [34]. PCR products were purified and directly sequenced in both directions. To minimize the possibility of amplification errors, high-fidelity Taq was used. The internal primers used for sequencing were 18S R536 [5′-CTGGAATTACCGCGGCTG-3′] and 18S R1052 [5′-AACTAAGAACGGCCATGCA-3′].

Finally, we used a metabarcoding approach, trying to target the largest number of eukaryotes. The DNA extracted with proteinase K was amplified using the Euka02 primers (Forward: TTTGTCTGSTTAATTSCG; Reverse: CACAGACCTGTTATTGC) which are highly generalist primers that amplify most of eukaryotes with limited bias [35,36]. These primers amplify a short (~120 bp) region of the 18S rDNA (V7), and are thus suitable for the analysis of degraded or poor quality DNA [36,37]. DNA was amplified following the same protocol of [37] and sequenced using an Illumina Hiseq 2000 platform (see [37] for complete details on sequencing and bioinformatics processing). The detected sequences were taxonomically assigned using the Ecotag program of the OBITOOLS package, on the basis of NCBI database [38]. We analysed two DNA aliquots, each with four PCR replicates. We also ran 9 PCR controls, each replicated four times, including PCR mix but no template DNA to identify environmental contaminants [39]. To avoid the risk of false positives we only considered taxa detected in >50% of PCR replicates with >10 reads [40].

### 2.8. Ethics

The study design, the samplings and the surgery on the fire salamanders was approved by the ethical committee of the Lombardy Region Authority and was authorized as complying with the regional law 10/2008, permission number: T1.2016.0052349. After the removal of the cysts and recovery, each individual was released in the exact place of its collection.

## 3. Results

### 3.1. Cysts Occurrence in Fire Salamander Populations

The cyst’s occurrence was first detected in 2013 in two males out of 66 salamanders observed. One of the males showed two turgid swellings occurring in a median position at the throat level, one left and one right separated by less than a millimetre (Figure 1); the cyst diameter was 10 and 8 mm respectively. The cyst size varied among individuals, while its position was always similar. We did not detect signs of external damage or injuries at the throat level, but on the cysts some small points (2 and 1 respectively) characterized by very thin skin stratus and appearing of a light black colour, were visible. The number of individuals detected per sampling in Fontana del Guercio locality, varied from five in a sampling time in 2017 to 202 during a sampling period in 2018 (Appendix A). From 2014 to 2020 the proportion of affected individuals was on average (±SE) 18.5% ± 8.1% per year. Both the proportion of individuals displaying the pathogen and the total number of adult salamanders detected along the transects varied across the years of monitoring (Appendix A); on average, in the Guercio protected area, considering the different samplings and transects, the largest proportion of affected salamanders was observed during 2015. In most cases, salamanders showed one single cyst, but 12.09 ± 6.08% of the affected individuals per year showed two of them.

In 2016, we observed for the first time, an affected adult in the transect performed in the second locality. Here, since 2016 the percentage of affected salamanders has been on average 1.18 ± 0.4%.

### 3.2. Histological Analysis

Macroscopically, cyst masses were spherical, located on the ventral surface of the jugular region, frequently bilateral, and, once removed, ranged between 0.2 cm and 0.4 cm in size (Figure 1A,B). They were histologically composed of a thin capsule encircling a central area of mucus material admixed with numerous granulocytes (morphologically compatible with heterophils), and a lesser number of plasma cells and lymphocytes (Figure 3D,E). Embedded in the mucus, numerous ciliated protists were detected (Figure 3F). A final diagnosis of a heterophilic cyst with intralesional microorganisms was made.

### 3.3. In Vitro Isolation

In cyst cell cultures, different cellular phenotypes were observed. Most of them were recognized as salamander leukocytes, mainly represented by granulocytes, among which peculiar cells were always present (Figure 3A). These latter appeared as spherical protist-like cells of about 30 µm in total length. Their cytoplast was generally heterogeneous and the cell membrane was covered by numerous motile cilia/undulipodia (Figure 3B,C).

### 3.4. Electron Microscopy

Electron microscopy analyses provided a detailed description of the protist-like cells in the salamander cysts. They appeared as unicellular organisms with a diameter of 10 μm, characterized by numerous undulipodia (Figure 4A,F,G). These structures protruded from the cell body (Figure 4B–D) and completely covered the cellular membrane (Figure 4H). Undulipodia were as long as the cell body, about 10 μm; together, the cell body and the undulipodia constituted what appeared as an unicellular organism of 30 μm in total length (Figure 4A,H). The cytoplasm was rich in electron dense and transparent inclusions (Figure 4A). At higher magnification, it was possible to appreciate the exogenous materials contained in the vesicles (Figure 4B). Mitochondria with peculiar cristae were also observed (Figure 4E).

### 3.5. DNA Analyses

None of the several DNA amplifications and sequencing attempts clarified the identity of the organisms. Using genomic DNA extracted from whole cyst samples as a template, no rDNA fragment was amplified employing either protists primers [29] or protozoan-specific ones [30,31,32]. DNA amplification from in vitro cell isolates was also not successful. Sequencing analysis of the only amplified DNA fragment revealed the presence of fungi belonging to the genus *Cladosporium*, which was probably due to contamination. Using metabarcoding, we detected only one molecular taxonomic unit in >50% of PCR replicates (sequence: ctcaaacttccatctactaaacgtagatagtccctctaagaagccaaaaaagccaaccaaagtcgacccggctatttagcaggttaaggtctcgttcgttat). The Ecotag program assigned this sequence to the fungal genus *Meira* (Brachybasidiaceae). However, this fungus was also detected in 25% of controls, suggesting that it is a contaminant.

## 4. Discussion

The combination of extensive field surveys with molecular and morphological analyses allowed us to detail a new disease affecting one of the most widespread amphibians in Europe. Morphological analyses showed that the turgescence was composed mainly by mucus encapsulating numerous protist-like cells, which appeared to be embedded in small niches among the tissue. The main feature of these cells was the presence of long peculiar cilia similar to undulipodia covering the cell membrane. In the cyst, numerous leukocytes of the salamander were also recognizable, suggesting an active immune response. In vitro studies confirmed the morphology of the cells and provided more information about their cilia/undulipodia which were motile. Electron microscopy further characterized the cells. Their heterogeneous cytoplasm rich in digestive vacuoles strongly suggested phagocytosis activity. However, a definitive diagnosis remained elusive because morphological and ultrastructural data did not allow identification of the cells encapsulated by the cysts, as we were unable to match the morphology of these cells with a known pathogen.

Amphibians are affected by a large spectrum of parasites and pathologies [11,41,42] and are parasitized by multiple protists [41]. Initially we supposed a similarity between the encapsulated microorganisms and some mesomycetozoan stages [23] that parasitize fish; however, they had a different position and shape of their flagella [23]. This similarity led us to perform molecular investigations following [29], which reported for the first time the occurrence of *Amphibiocystidium* sp. in the red-spotted newts. Despite some similarities with *Amphibiocystidium*, we failed to detect its DNA and, even though we attempted multiple PCR primers and techniques, we only amplified the DNA of contaminants. The mucus of cysts could inhibit DNA extraction or amplification by reducing their efficiency [43,44], still a modification of the DNA extraction protocol according to [28] did not provide significant improvements. The absence of DNA amplification could be because, even if the cysts may attain conspicuous sizes, the encapsulated cells were quite sparse, possibly limiting the amount of DNA. Furthermore, the amplification success of primers can be highly heterogeneous across taxa [37]. Even if we employed both specific and universal primer combinations, it is possible that the potential pathogen has mismatches in the priming region that hamper amplification. In principle, the location of the cysts could also be suggestive of internal disorders linked to production of salamander cells with abnormal morphologies. Nevertheless, we can consider neoplastic formations as unlikely because the encapsulated cells are completely different from described amphibian cells and the encapsulation reaction clearly seems to underline an immune reaction toward an external agent. Ciliated amphibian cells have been recorded only in the gills of some urodeles, but they show very different shapes and features compared to the detected ones [45,46].

## 5. Conclusions

Despite the numerous morphological, ultrastructural, and molecular analyses, we failed to unambiguously identify the cause of the disease. The significant frequency of salamanders showing cysts and the features of the cells encapsulated by those cysts, suggest that this disease could be caused by a slow growing and enigmatic wildlife pathogen.

Our study suggests that this disease could be caused by a pathogen that might be an unidentified protist; if dispersal by infected salamanders is likely to become a threat for the surrounding populations, the occurrence of other possible vectors should not be excluded. Waterways are considered highly suitable for the survival and spread of several pathogens [47]: a stream flows down from the Guercio protected area and connects with a complex hydrographic network in the basin of the river Lambro. Moreover, the Guercio protected area has numerous visitors each year, represented mainly by local people but also including schools and even people from abroad. Efforts will thus be necessary to make all the stakeholders of the area, from wildlife managers to visitors, aware of the presence of the phenomenon and of the caution measures to adopt to avoid its spreading. Moreover, we hope the findings of our study stimulate further studies, both on the biology of the new discovered cysts and on the occurrence of other potential animals affected, from fishes to other amphibian species.

## Figures and Tables

**Figure 1 animals-12-00696-f001:**
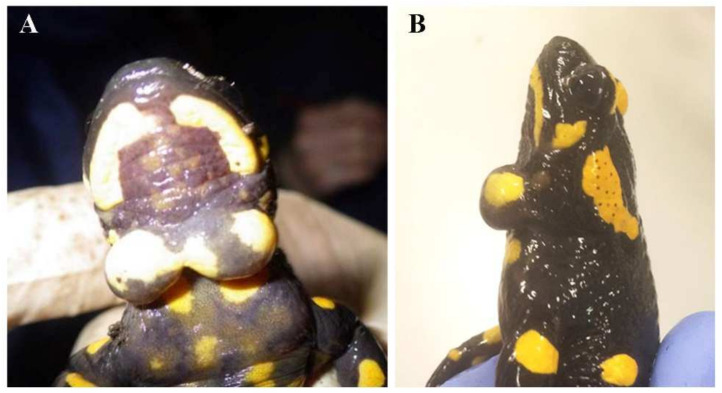
Salamander cysts. Ventral (**A**) and lateral (**B**) view of a male *Salamandra salamandra* with two turgid cysts occurring in a median position at the throat level.

**Figure 2 animals-12-00696-f002:**
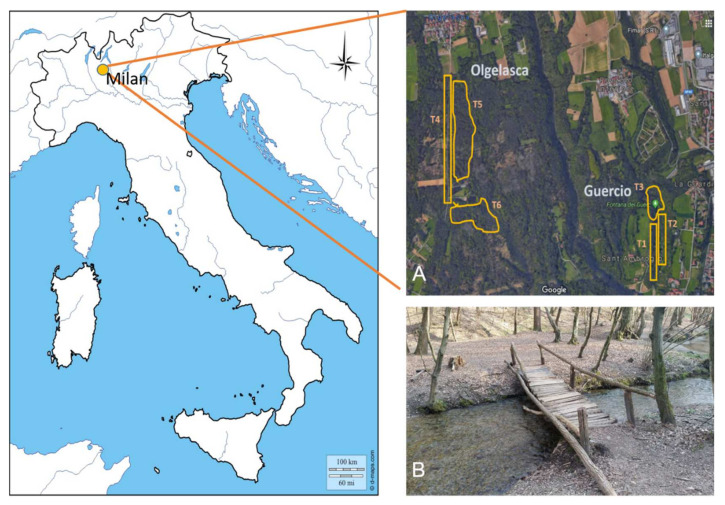
Location of the study area. (**A**) location and area of the transects monitored, identified by the letter T; “Guercio” identifies the main locality on the regional protected area named “Riserva del Guercio” in which the pathogen was first detected. “Olgelasca” identifies the adjacent locality in which the pathogen appeared in 2016. (**B**) picture of the protected area between transect 1 and transect 2.

**Figure 3 animals-12-00696-f003:**
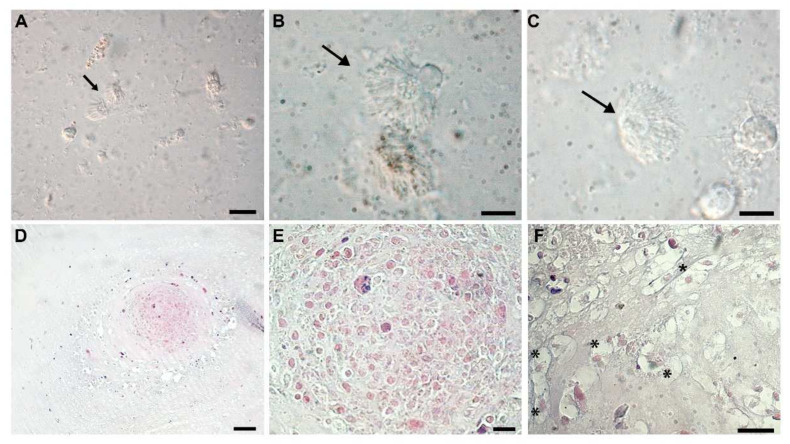
Morphological analyses. (**A**–**C**) Cyst cell culture in which peculiar ciliated cells are observable (arrow). These protist-like cells displayed heterogeneous cytoplast and their cell membrane was covered by numerous motile cilium-like structures (**B**,**C**). (**D**–**F**) Histological analysis revealed that the cyst masses consisted of a thin capsule of connective tissue encircling a central area of mucus material in which numerous granulocytes, plasma cells, and lymphocytes were detected (**D**,**E**). Among these, ciliated protist-like cells were present trapped in the mucus (**F**); asterisks identify trapped cells. Scale bars: (**A**) 20 µm; (**B**,**C**) 10 µm; (**D**) 250 µm; (**E**,**F**) 50 µm.

**Figure 4 animals-12-00696-f004:**
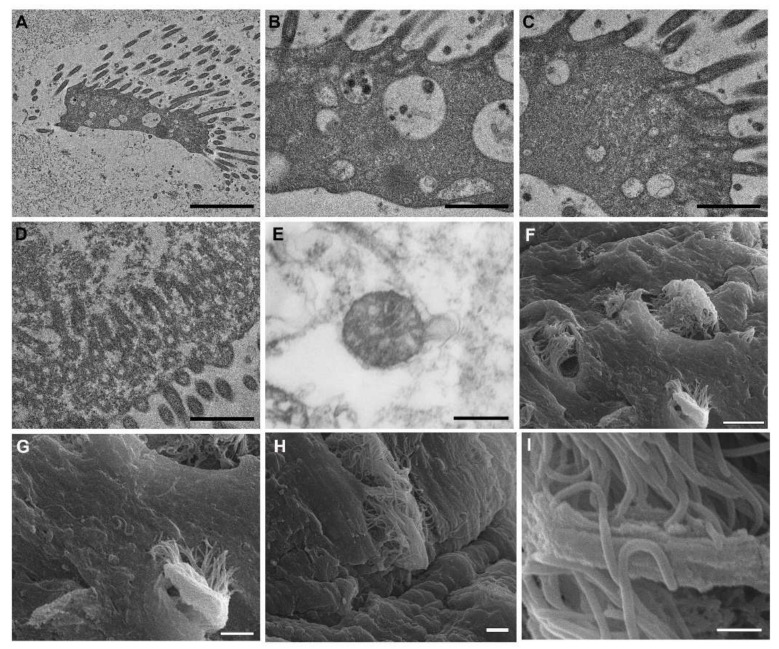
Electron microscopy analyses. (**A**–**E**) Transmission electron microscopy of the pathogen. In cyst mucus, numerous protist-like cells (**A**) with long undulipodia were found. At higher magnification (**B**), electron dense and transparent vacuoles were observed as well as undulipodium rootlets regularly distributed along cytoplasm periphery (**C**,**D**). Mitochondria with peculiar cristae were detected (**E**). (**F**–**I**) Scanning electron microscopy. Trapped inside mucus niches, protist-like cells were observed (**F**,**G**). Their main feature was the numerous and long cilia/undulipodia which covered the cell membrane (**H**,**I**). Scale bars: (**A**) 2 µm; (**B**–**D**) 1 µm; (**E**) 500 nm; (**F**) 10 µm; (**G**) 5 µm; (**H**) 2 µm; (**I**) 1 µm.

## Data Availability

Not applicable.

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
