# Peer review of "A New Disease Caused by an Unidentified Etiological Agent Affects European Salamanders"

_animals, 2022, doi:10.3390/ani12060696_

Round 1

Reviewer 1 Report

This paper present important finding of  new infestation in Salamandra salamandra populations and provide the morphological description of the disease and pathogen. The research is well performed and presented. I hope the authors will continue survey of affected populations and will collect more data.

Line 17. Reformulate the sentence. Replace “were undescribed just 25 years ago” with “described recently” or “have been described during last two decades”.

Line 17.  Delete “ unforeseen”

Line  19  Delete “ which existence is not reported in amphibians yet.”

Line 19 “ During these years” – please define the period

Line 35. “ about 10 @m long” – different number is reported in line 23

Line 77.  Delete “for the first time”.

Line 113, 221. What precaution measures were taken to prevent spreading the disease by handling salamanders during survey?

Line 249. Please reorder the micrographs according to magnification. What the asterisk are about?

Line 257. The macroscopic description of the cyst (Paragraph 3.3) should be given before Paragraph 3.2.

Figure 3 and Figure 4 are identical. 

Line 296. What would be “epidemiological surveys”? Please rephrase

Line 297. Please reformulate. Replace “an unprecedented phenomenon” with "a new disease".

Line 352. The link does not work.

During your survey, have you had recaptures of individuals with cysts? It would be important to know the survival rate of infected individuals in the future.

Lines 298,299.  Delete or reformulate the sentence. Suggestion: Morphological analyses showed that the turgescence ….

Lines 300,301. It is not clear “The main feature of these cells was the presence of long peculiar cells similar to  undulipodia covering the cell membrane” How is the main feature of  cells thes presence of cells?

Line 305. “their cilia/undulipodia  which seemed to beat synchronously” – I can’t see that in the results.

Lines 333-337. This sentence is hard to follow. Please reformulate.

Reviewer 2 Report

Dear Authors, so interesting paper and high important to protect this species in Europe. I'm curious how many other researchers will be inspired after reading this paper to focus on speciments of salamandra. I hope publishing this paper will open for you many opportunities for cooperation with other scientists in Europe on this topic.

I have just a few remarks, that I describe by lines:

5 - I'm not sure what 'F' at the end means?

39 - I would add among key words also name of the species: fire salamander

50 - it should be: well-being

58 - without "the" pathologists, you can confirm it by native speaker

77 - I would add to Latin name: (..... L. 1758)

96 - "in the north-eastern part of its range" - you mean  whole range of the species in Europe? Here: Ukraina and Romania? 

114 - Figure 2: A map. I don't understand the idea of orange lines on the map. In the text there was not written that research areas are located e.g. 1 km from  administrative borders of the Milano city... I would rather don't show map of whole Italy, but show only North part of the country with localization of both areas on the background of Milano... or maybe regions, you mentioned Lombardy region...

142 - there is a mistake. You wrote: ....of the culture medium was carried out every 5 days. It should be: every day, each day, or all 5 days.

145-146 - try not to break 24 h into two lines.

155: for two hours, the same like in 195 line

210 - (see [37] PUT A SPACE for...)

268 - Fig. 4 H - I don't see this mentioned figure

279, 280, 281 - where are figures 4: I, G and H ? I see that figures 3 and 4 are the same. You need to add proper ones. 

311 - there are no needed characters, like:  {Densmore, 2007 #4209;Lunghi, 2018 #2271;Rollins-Smith, 2017 #4182}. Moreover, why you did not put numbers in [ ]? 

333, 335 and e.g. 341 and whole text - try not to put single letters at the ends of lines. It is a basic editing rule, in each language.

357 - RM, instead of RM:

361 - two dots, instead of one.

373 - Conflict of Interests should start from a new line. Here you copied official statements, probably by mistake - it need to be corrected. Match the content to your situation. 

410, 413, 415/416, 430, 432, 456, 458, 465, etc. in whole "References" chapter - you should put short names of Journal like in other positions.

461 - repetition of 2015 year two times

466-468 - check correct reference. It is not like you written:  (vol 10, e0139633, 2015). Plos One 2016, 11, (4)

470 - 2018 instead of 2018

490 - no need "1".
